# Bleeding Complications in COVID-19 Critically Ill ARDS Patients Receiving VV-ECMO Therapy

**DOI:** 10.3390/jcm12196415

**Published:** 2023-10-09

**Authors:** Armin Niklas Flinspach, Dorothée Bobyk, Kai Zacharowski, Vanessa Neef, Florian Jürgen Raimann

**Affiliations:** Goethe University Frankfurt, University Hospital Frankfurt, Department of Anaesthesiology, Intensive Care Medicine and Pain Therapy, Theodor-Stern Kai 7, 60590 Frankfurt am Main, Germany

**Keywords:** critical care, acute respiratory distress syndrome, severe acute respiratory syndrome coronavirus 2, extracorporeal membrane oxygenation, transfusion, bleeding, anticoagulation

## Abstract

Veno-venous extracorporeal membrane oxygenation (VV-ECMO) therapy is rapidly expanding worldwide, yet this therapy has a serious risk of bleeding. Whether coagulation-activating viral infections such as COVID-19 may have an impact on the risk of bleeding is largely unknown. This study conducted a monocentric investigation of severely affected COVID-19 patients receiving VV-ECMO therapy with regard to the occurrence and possible influences of minor and major bleeding and transfusion requirements. Among the 114 included study patients, we were able to assess more than 74,000 h of VV-ECMO therapy. In these, 103 major bleeding events and 2283 minor bleeding events were detected. In total, 1396 red blood concentrates (RBCs) were administered. A statistically significant correlation with the applied anticoagulation or demographic data of the patients was not observed. Contrary to the frequently observed thromboembolic complications among COVID-19 patients, patients with VV-ECMO therapy, even under low-dose anticoagulation, show a distinct bleeding profile, especially of minor bleeding, with a substantial need for blood transfusions. COVID-19 patients show a tendency to have frequent bleeding and require repeated RBC transfusions during VV-ECMO. This fact might not be solely explained by the mechanical alteration of ECMO or anticoagulation.

## 1. Introduction

Extracorporeal membrane oxygenation (ECMO) entered critical care medicine in the 1970s and has gained increasing importance, especially in the recent decades and even more during the coronavirus pandemic (COVID-19). As a way to maintain adequate oxygenation and decarboxylation of critically ill patients with severe pulmonary compromise, tremendous progress has been achieved [1,2]. While veno-venous ECMO (VV-ECMO) is used for treating a severely compromised gas exchange, the replacement of cardiac function by ECMO is also becoming increasingly common in the sense of a circulatory supplementation technique by veno-arterial ECMO (VA-ECMO).

Among the first problems of membrane oxygenation were rheological difficulties, such as leakage of plasma at the oxygenator membrane and thrombotic adhesions, sometimes with fatal device failures [3]. Due to the large xenogeneic surface, coagulation activation is inevitable, which necessitates appropriate anticoagulation along with ECMO therapy. This anticoagulation is routinely performed with unfractionated (UFH) or fractionated heparin (low-molecular-weight heparin; LMWH) or direct factor Xa inhibitors, depending on the in-hospital standard and valid recommendations. The performance of extracorporeal membrane oxygenation without anticoagulation remains a distinct rarity and is not recommended in anticipation of a prolonged run [4]. Major technical advances in oxygenator surface, circuit design, and cannulas, such as (heparin) coatings, helped this technique to become more widespread and tolerable, including decreasing complications. However, bleeding remained a relevant problem during ECMO treatment [5,6]. Reflecting the highly invasive nature of this technique, a still high mortality rate of more than 50% is still evident despite all advances mentioned above [7]. Among the main causes, in addition to failure of organ recovery, are bleeding complications, which are mainly based on the need for therapeutic anticoagulation.

Previously, ECMO implementation was reserved for dedicated expert centers and unknown in the public perception until a few years ago. The COVID-19 fundamentally changed this [8]. Severe COVID-19-associated acute respiratory distress syndrome (CARDS) requiring ventilation already showed a considerably prolonged duration of mechanical ventilation [9]. Correspondingly, it became apparent very early that, analogous to conventional ventilation, it often required prolonged ECMO runs as well. A meta-analysis showed an average treatment duration of 13.9–15.1 days, but in addition, smaller case series with durations of more than 100 days have been published [8,10,11,12,13,14,15,16]. Due to the high demand, a corresponding accelerated diffusion of ECMO endpoints and their applications became apparent. This development showed a further significant acceleration, although it took place in the last decade in Western countries. An increase in treatments of 200–400% was detectable [7,17].

However, COVID-19 also showed numerous end-organ compromises in addition to pulmonary involvement; of particular intensive care significance are thromboembolic events, sedation aging, and renal failure [18,19]. Thus, an impairment of coagulation in the sense of procoagulability with accompanying partly deleterious thromboembolic events in the sense of pulmonary artery embolisms became apparent early. This led to a long-lasting discussion about the optimal anticoagulation therapy for hospitalized COVID-19 patients [20]. For a long time, no clear recommendations existed for the anticoagulatory treatment under VV-ECMO, which resulted from the tension between a procoagulatory disease and an equally therapeutic anticoagulation [21].

However, a clear description of how often bleeding complications occurred during corresponding ECMO runs, what their characteristics were, and their consequences for Hemoglobin (Hb) loss and transfusion requirements has scarcely been described. Furthermore, the extent to which the bleeding had relevant therapeutic effects and which concomitant circumstances caused them is far not well investigated for the special case of COVID-19 patients.

## 2. Materials and Methods

We conducted a retrospective monocentric observational study at a tertiary university hospital. The study protocol was checked and approved by the institutional review board (IRB) (#20-643). Due to the retrospective design, a waiver of written informed consent was approved by the IRB. The study was conceived and designed in accordance with the actual recommendations of the Declaration of Helsinki [22]. A subset of the patients included in this study had already been included in other studies of the COVID-19 cohort treated by us (e.g., analysis of sedation behavior and motility during VV-ECMO treatment) [16,18,23].

### 2.1. Patient Population

Within the observation period from 15 April 2020 to 31 March 2022, all adult patients (>18 years) who had been treated with VV-ECMO were included in the study. The ARDS study center operates with cardiohelp^®^, rotaflow I and II^®^ devices (Getinge AB, Gothenburg, Sweden) for VV-ECMO, and an Elisa 800^®^ (Löwenstein Medical, Bad Ems, Germany) or otherwise a Hamilton G5^®^ (Hamilton Medical, Bonaduz, Switzerland) intensive care ventilator for mechanical ventilation.

Patients with primary COVID-19 disease over the age of 70, as well as with pre-existing life-limiting medical conditions (e.g., cancer, Parkinson’s disease, dementia, etc.), lung disease requiring therapy (e.g., pulmonary fibrosis, advanced chronic obstructive pulmonary disease, or severe bronchial asthma), or advanced cardiovascular disease, were excluded from therapy with VV-ECMO according to the ELSO guideline recommendations and internal standard operating procedures [8]. Due to the existing therapy with invasive ventilation via an endotracheal tube and prone positioning, previous therapy with oral anticoagulants was discontinued and switched to unfractionated heparin. Existing therapies with antiplatelet agents were critically evaluated and most of them were discontinued. If bleeding events occurred in possible association with an ongoing effect of antiplatelet drugs, we quantified these effects using a multiplate^®^ analyzer (Roche, Basel, Switzerland) for point-of-care diagnostics. The corresponding POCT results were considered in conjunction with the absolute platelet count and the dynamics of any equally present thrombocyte drop due to different reasons in order to optimize coagulation.

All patients received a comparable mechanical ventilation under VV-ECMO therapy based on pre-existing recommendations for lung protective ventilation with a target tidal volume of ≤6 mL/kg ideal bodyweight, while maintaining the lowest possible peak pressure, as well as a driving pressure of <15 mbar. In addition, therapy was based on standardized intensive care therapy, and ECMO treatment during VV-ECMO therapy was conducted according to appropriate available current guidelines and in-house recommendations [24]. Regarding COVID-19-specific therapeutic strategies, treatment was based on the current recommendations at that time [25,26,27,28]. During ECMO treatment, the function of the oxygenator was monitored by measuring the maximum oxygenation capacity daily (within 12 L per minute of 100% oxygen gas flow and resulting p_a_O_2_). Platelet count and lactate dehydrogenase (LDH) were monitored to assess mechanical deterioration of cellular blood components. In suspected cases of hemolysis, free hemoglobin and haptoglobin were determined during the course of therapy, for closer quantification [29]. In the context of invasive therapy with VV-ECMO, a target hemoglobin value of 9 g/dL was targeted in hospital protocols to optimize the availability of systemic oxygen (D_a_O_2_ increase). To monitor anticoagulation with UFH, the activated partial thromboplastin time (aPTT) was measured every four hours and, in the case of bleeding, the cause and transfusion were documented. The aPTT was determined to guide continuous anticoagulation with unfractionated heparin, intending for a target aPTT between 40 and 60 s. A more precise target within this range was determined on an individual case basis depending on the actual ECMO blood flow and any coexisting bleeding events. The attribution of red blood concentrates to a specifically designated bleeding event occurred if it was transfused no more than 24 h after bleeding had occurred.

### 2.2. Data Collection

Clinical data were collected continuously using a patient data management system (PDMS; Metavision 5.4, iMDsoft, Tel Aviv, Israel) and laboratory system (Lauris V.2.23, Nexus Swisslab, Berlin, Germany). We recorded demographic data, ECMO parameters (liters per minute (LPM), rounds per minute (RPM), sweep gas flow, oxygenator changes), laboratory results (aPTT, Hb, p_a_O_2_, p_a_CO_2_, LDH), transfusions, and bleeding events.

The documentation of major bleeding was based on established criteria in the case of symptomatic bleeding in a critical area or organ, such as intracranial, retroperitoneal, or intramuscular with compartment syndrome, and/or bleeding causing a decline in Hb of ≥2 g/dL or leading to the immediate transfusion of two or more RBCs [30]. For analysis purposes, values of a measurement parameter were input into Excel (Microsoft Excel 365, Redmond, WA, USA) for every eight hours of treatment.

### 2.3. Statistical Analysis

No statistical power calculation was conducted prior to this study, due to the retrospective study design. The sample size was based on the available data of a maximum number of patients included according to the inclusion criteria. The categorical variables are presented as counts and percentages. Not normally distributed variables are described as medians (interquartile range, IQR (25/75)). Demographical patient data and clinical differences between the bleeding groups were assessed using Fisher’s exact test for categorial variables and the Mann–Whitney U test and Kruskal–Wallis test for continuous variables, as appropriate.

The statistical tests applied were all two-tailed, and results with *p* < 0.05 were considered to be statistically significant. All calculations/analyses were performed with SPSS^®^ (IBM Corp., Version 26, Chicago, IL, USA).

## 3. Results

During the observation period, 21,781 patients were admitted to one of the existing monitoring wards at our tertiary university center. Among these were 664 patients with moderate-to-severe COVID-19-associated ARDS as defined by the BERLIN criteria. Of these, 114 patients received VV-ECMO therapy based on their disease severity, of whom all were included in the study (Figure 1).

Diagram of the study screening and selection process for study inclusion.

In accordance with the invasive therapy and the unclear chances of success, especially at the beginning of the COVID-19 pandemic, a median age of 54 years was treated. Further demographic data can be found in Table 1.

Among the different groups presented in Table 1 with respect to detected bleeding events, no statistically significant differences could be found.

### 3.1. Bleeding Events and Assignment

In association with the treatment with VV-ECMO, 103 major bleeding events in 23 patients were identified in the medical record. These could be subdivided into ten intracranial hemorrhages, all of them with an unfavorable prognosis and end of treatment. Seven gastrointestinal hemorrhages and three intrapulmonary hemorrhages required intervention, all of which were difficult to control with interventional therapy. Multiple major bleedings events of the same etiology were repeatedly observed in the same patients, such as gastrointestinal hemorrhages and intrapulmonary bleedings.

In the context of cannulation to the device or the cannulae themselves, eight severe bleeding events were documented. In total, 20.2% of all VV-ECMO patients experienced a major bleeding event during the course of therapy.

In addition to the major bleeding events, minor bleeding was detected at 2283 measurement points, affecting 56 patients (49.1%). During the course of therapy of the CARDS patients, repeated severe nosebleeds without connection to external manipulations were observed, which regularly required oropharyngeal tamponade to stop the bleeding. These events, as well as persistent oozing bleeding after tamponade, account for the majority of minor bleedings. Among the 2283 bleedings, 17 bleedings from the urinary bladder after bladder catheterization were documented, with an exception concerning only one patient after a traumatic initial placement, as well as 184 small bleedings at the ECMO cannulation insertion sites, especially after placement of the cannulae. In the context of VV-ECMO therapy, two patients received antiplatelet medication for short intervals until a gradual decrease in platelet counts led to the discontinuation of this therapy without a correlation to a bleeding episode being observed. In total, 2082 measurement points could be assigned to bleeding from the nasopharynx.

### 3.2. Transfusion Therapy

Within the analyzed 74,792 h of VV-ECMO therapy, documentation of minor bleeding was identified at 2283 of 9349 (24.4%) measurement points.

To maintain hemoglobin levels under therapy, 1396 RBCs were transfused. The majority of RBC transfusions could be assigned to the observed minor bleeding events (see Figure 2A), and transfusion was mainly (84%) performed as a single administration (see Figure 2B).

Figure 2 shows the pie chart illustration of all transfusions administered to included patients during VV-ECMO treatment. A shows the percentage of transfusions attributed to minor bleeding events, major bleeding events, or without context to bleeding events. B shows the number of red cell concentrates administered in a measurement interval.

Statistical analysis revealed that the aPTT present during or at the bleeding event had no effect on the occurrence of a bleeding event per se or the likelihood of transfusion (Figure 3). A total of 110 oxygenator changes were monitored during the observation period, resulting in a total use of 224 membrane oxygenators. The average shelf life of an oxygenator was thus 13 days and 22 h.

Figure 3 shows that the activated partial thromboplastin time in seconds compared between minor and major bleeding, while on continuous unfractionated heparin and present at the onset of minor or major hemorrhage, does not indicate a significant difference. Abbreviations: aPTT, activated partial thromboplastin time; s, seconds.

## 4. Discussion

We were able to analyze bleeding events of 114 VV-ECMO runs covering nearly 75,000 h of treatment with more than 9000 measurement points. There was a high incidence of minor bleeding events, which correlates with the incidence found in the literature among critically ill patients [3,30,31].

However, a substantial proportion of transfusions without chronological correlation (≥24 h) to minor or major hemorrhage were equally found. The high likelihood of transfusion, judging from previous data, is due to a mixed pattern of COVID-19-associated impairment of iron metabolism and associated hematopoietic activity and the ECMO run itself [32,33]. VV-ECMO therapy causes a considerable amount of hemolysis due to the large contact area and mechanical alteration in the centrifugal pump of corpuscular blood components [34]. The unfortunate coincidence of COVID-19-related impaired hematopoiesis when in critical condition with a parallel increased turnover due to therapy (blood sample collections and VV-ECMO) is to be interpreted as the cause of the observed need for transfusion in absence of a bleeding event [35].

The majority (61%) of the transfusions performed should be seen in a chronological context related to minor bleedings detected at 2283 measurement points. The majority of these appear to be hemorrhages from the nasopharynx which were not caused by manipulations (e.g., gastric tube insertion, Wendel tubes, etc.) during the course of treatment [36,37]. It remains unclear to the attending intensivists as well as the ear, nose, and throat (ENT) specialists, who were repeatedly involved as consultants, where these very frequent and persistent diffuse hemorrhages of the mucosa originated from [37].

Major bleeding was detected in more than every fifth patient (20.2%) and was often unfavorable. Thus, all ten patients in whom intracranial hemorrhage was detected died under VV-ECMO therapy due to the extent of the hemorrhage and the unanticipated survival based on the neuroimaging findings. Accordingly, in our cohort, we observed an overall mortality of 65.8%, which is in line with other published studies [14,31]. In the case of observed major bleeding, an increased mortality of 87.0% was found. Accounting the 93 otherwise nonintracranial hemorrhages detected, in terms of gastrointestinal-, pulmonary-, or cannula-related bleeding, these caused in total the substantial transfusion requirement of 5% of all RBCs administered.

One reason for the large amount of 1396 RBCs administered during the observation period may be explained by the intended target Hb value under VV-ECMO therapy to increase the DaO_2_; given the critical oxygenation and decarboxylation impairment of these patients, we favored a Hb of ≥9 g/dL. Even though recent analyses postulate that even lower Hb levels may be sufficient for adequate oxygenation under VV-ECMO, our approach is in line with current therapy standards [38,39,40]. Many of the patients with CARDS required complete VV-ECMO oxygenation and decarboxylation via the device due to an almost complete inability of the lungs to sustain a relevant gas exchange over long treatment periods. In this collective of patients, the attending physicians felt compelled to use all possible means to maintain sufficient oxygenation in terms of SaO_2_. Accordingly, the maintenance of a sufficient hemoglobin value played a significant role and the already mentioned value of more than nine was intended. Even though repeated derailments of coagulation under continuously applied unfractionated heparin via aPTT have been documented in the cohort we studied, no association with bleeding events could be found. Anticoagulation was not adjusted to a therapeutic level in the majority of cases due to the repeatedly seen difficult-to-control bleeding from the nasopharynx as shown in Figure 3. Especially in combination with the procoagulant events repeatedly described for COVID-19, the frequency of bleeding on the one hand and the relatively low number of necessary oxygenator changes on the other hand is astonishing, because internationally as well as in our cohort, prolonged courses of therapy were observed in the majority [11,16,41]. A pre-existing critical COVID-19 condition with concomitant endotracheal intubation and prone positioning therapy prior to VV-ECMO application allowed for the timely discontinuation of all oral anticoagulants as well as the majority of pre-existing medications with thrombocyte aggregation inhibitors. All suspected cases of medication-related bleeding were critically reviewed using point-of-care diagnostics. It remains the still not fully understood alteration of the coagulation cascade in critically ill COVID-19 patients, especially in the light of the considerable impairment of coagulation by the maintenance of an extracorporeal circulation.

### Limitations

As a retrospective single-center study, the project we conducted has a corresponding limitation with regard to transferability to other centers. This resulted in data collection and analysis limited to the designated 114 cases that we were able to retrospectively collect. The frequency of a POCT-supported consultation of the rheological situation or the consequences drawn thereupon cannot be traced in detail due to the lack of digital documentation. By a solely in-house observation, we are unfortunately not able to report a more differentiated picture with regard to possibly existing long-term impairments in connection with the occurred bleedings, especially from the nasopharynx [42]. Similarly, we cannot provide information on the severity of postintensive care syndromes and associations with bleeding events, if any, that occurred in the cohort [43]. Both issues might provide important information on the influence of bleeding events on long-term outcome and thus should be the subject of future investigations.

## 5. Conclusions

Life-saving, albeit highly invasive, therapy with VV-ECMO is associated with a high risk of bleeding complications in our collective of CARDS patients (69.3%). Similarly, these patients frequently require red blood cell transfusions during therapy with VV-ECMO, which does not seem to be explained by anticoagulant therapy alone. Alternative causes of the need for transfusions, such as mechanical alteration by the oxygenator and centrifugal pump, require further investigations.

## Figures and Tables

**Figure 1 jcm-12-06415-f001:**
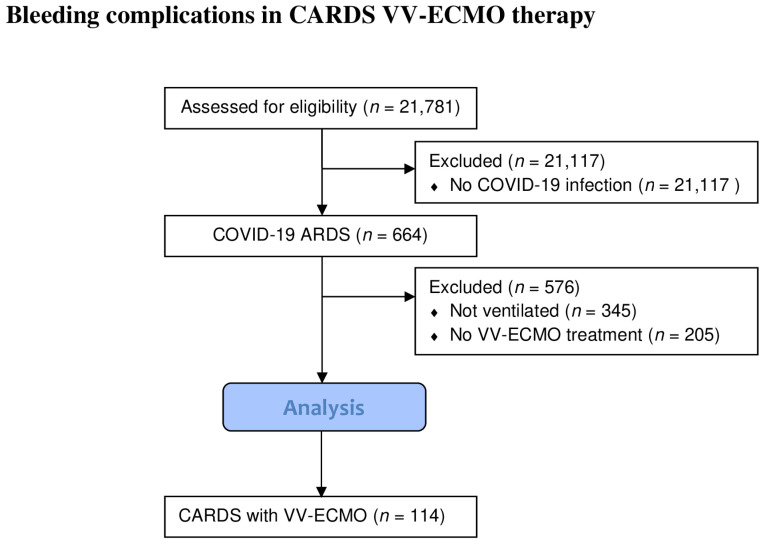
Patient Screening.

**Figure 2 jcm-12-06415-f002:**
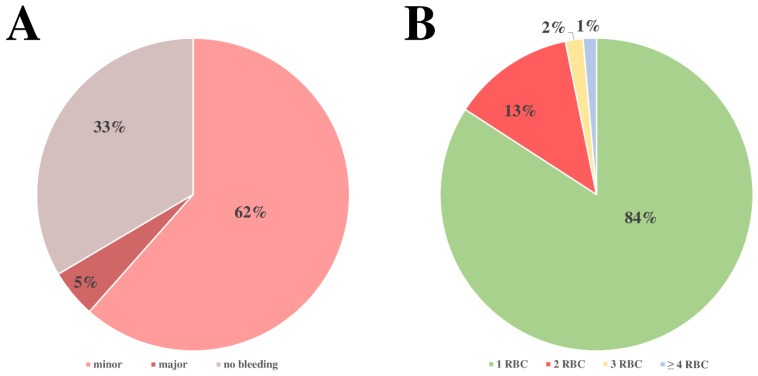
Allocation of blood transfusions to bleeding events and quantity of RBCs transfused.

**Figure 3 jcm-12-06415-f003:**
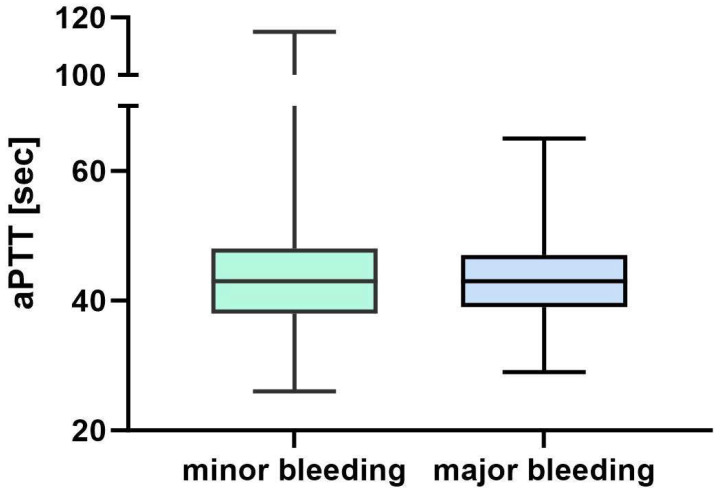
aPTT level at the onset of hemorrhage.

**Table 1 jcm-12-06415-t001:** Clinical characteristics of VV-ECMO CARDS patients.

	Major Bleeding *	Minor Bleeding	No Bleeding	All
Per patient *n* =	23 (20.2%)	56 (49.1%)	35 (30.7%)	114 (100%)
Overall	103	2283	0	2386
sex (male,%)	20 (87.0%)	45 (80.4%)	32 (89.5%)	97 (85.1%)
age (years)	52 (IQR: 38/66)	55 (IQR: 40/70)	52 (IQR: 36/68)	54 (IQR: 46/62)
diabetes mellitus	4 (17.4%)	17 (30.4%)	6 (17.1%)	27 (23.7%)
cardiovascular disease	0 (0.0%)	6 (10.7%)	3 (7.9%)	9 (7.7%)
chronic renal failure	0 (0.0%)	3 (5.4%)	2 (5.3%)	5 (4.3%)
arterial hypertension	8 (34.8%)	21 (37.5%)	14 (36.8%)	43 (36.8%)
smoking	3 (13.0%)	7 (1.3%)	3 (7.9%)	13 (11.1%)
weight	92.0 (IQR: 74/110)	92.0 (IQR: 69/115)	96.0 (IQR: 86/106)	95.0 (IQR: 81/109)
BMI (kg/m^2^)	31.0 (IQR: 27/35)	29.8 (IQR: 21/38.6)	31.0 (IQR: 22/40)	30.5 (IQR: 23.4/37.6)
mortality	20 (87.0%)	33 (58.9%)	22 (62.9%)	75 (65.8%)

Clinical characteristics and allocation according to minor/major bleeding of the included patients receiving VV-ECMO therapy. Patient characteristics are presented as median (± the interquartile range (IQR)) or as number (percentage) where applicable. Abbreviations: BMI, body mass index; kg, kilogram; m, meters. * At least one major bleeding, independent of any minor bleeding events.

## Data Availability

The data cannot be shared publicly. The datasets generated and/or analyzed during the current study are not publicly available due to national data protection laws but are available upon reasonable request from the corresponding author or via the data protection officer of the University Hospital of Frankfurt (Datenschutz@kgu.de (www.kgu.de, accessed on 24 August 2023)).

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
