# Peer review of "Bleeding Complications in COVID-19 Critically Ill ARDS Patients Receiving VV-ECMO Therapy"

_jcm, 2023, doi:10.3390/jcm12196415_

Round 1

Reviewer 1 Report

In the manuscript "Bleeding complications in COVID-19 critically ill ARDS patients receiving veno-venous ECMO therapy. "the authors present the results of the retrospective monocentric observational study. Authors describe an incidence of bleeding events in 114 COVID-19 ARDS patients treated by veno-venous extracorporeal membranous oxygenation (VV ECMO). Although the topic is quite interesting, the reviewer should stress a few important shortcomings of the manuscript.

Major comments:

  1. The topic is not new as well. A few papers describing the incidence, etiology, pathophysiology, and consequences of bleeding events in Covid-19 VV ECMO were published. The presented manuscript provides no new or additional essential information.
  2. The quality of the manuscript is low, and some critical uncertainties should be mentioned:

- According to the description, Table 1 presents data "according to therapy duration ". To be honest, I cannot find any link between the presented data and the duration of therapy

- Line 160: "In total, 23.9% of all VV-ECMO patients experienced a major bleeding event during the course of therapy. "However, this number does not correspond to the data presented in Table 1 (Major bleeding: n= 23; 20.2%)! The number of major bleeding is mentioned on the value 23,9% again at line 221

- Line 196 (description to Figure 3) – sentence "Major bleeding events showed significantly (p < 0.0001) larger Hemoglobin (Hb) loss "has nothing to do with data presented in the picture, and this "surprising" finding is not supported by further data/numbers. Moreover, the authors claim that "The majority of RBC transfusions could be assigned to the observed minor bleeding events (see 176 Figure 2A) "(line 175). These statements are quite contradictory and are not further explained.

- Discussion, line 204: "However, a substantial proportion of transfusions without chronological correlation (≥24 hours) to minor or major hemorrhage were equally found ". I am sorry again, but I cannot find supporting data in the "Results. "

- Line 215: "The majority of these appear to be hemorrhages from the nasopharynx which were not caused by manipulations (e.g. gastric tube insertion, urethral catheter, etc.)." The reviewer would wonder if the manipulation with the urethral catheter could cause bleeding from nasopharynx

- Line 224 "The mortality of 65.8% found in our cohort is in line with other published studies, is related to the major bleeding (see all 221 – 228 lines), but mentioned mortality of 65.8% is related to the whole group

- Lines 246 – 249 "Authors should discuss the results and how they can be interpreted from the perspective of previous studies and of the working hypotheses. The findings and their implications should be discussed in the broadest context possible. Future research directions may also be highlighted." I wholeheartedly agree.

- The "Limitations "chapter does not reflect real limitations (number of patients, study design, etc.) except for the mentioned monocentric nature. Links to long-term nasopharyngeal complications and post-ICU syndrome are irrelevant

- Conclusions are mostly speculative but not supported by presented data

- Language is cumbersome, and many sentences are difficult to understand (if even)

Minor comments:

  1. Line 53: The abbreviation "(C) ARDS“ is not explained; moreover, in the text authors use „CARDS“
  2. Line 89: Machines used for mechanical ventilation are termed "ventilators “ instead of "respirators“.
  3. Line 92: "All patients received identically mechanical ventilation.“ Do the authors mean identical mode? Identical parameters?
  4. Table 1. Why is hypertension not viewed as a cardiovascular disease?
  5. Table 1. Statistics should be involved
  6. and many others, including grammar, incomprehensible sentences, and paragraphs

Very low quality

Reviewer 2 Report

These are my suggestions: "The major limitation of the study seems to be of a methodological nature. No inclusion/exclusion data are provided on patients with any history of bleeding diathesis; no information is provided on the use of drugs (antiplatelets? anticoagulants?). Furthermore, there is no comparison group of non-covid patients; therefore it is not possible to conclude that COVID-19 is a factor favoring bleeding".

Reviewer 3 Report

First of all, I would like to thank the authors for this intensive and laborious study. Although the rate of spread of COVID-19 in the world has decreased, understanding the bleeding complications of veno-venous extracorporeal membrane oxygenation therapy will also be useful in the treatment of other respiratory lung diseases. In this sense, the study is up-to-date and makes significant contributions to the literature.

Author Response

We would like to express our gratitude to the reviewer for the time and care taken to improve the manuscript we submitted and are extremely delighted with the kind feedback.

Reviewer 4 Report

Lines 102 and 105 indicate the concentration of hemoglobin in mg / dL, it should be g / dL

The topic is quite original and relevant, and the solution of the problem as well as a gap in this area, in my opinion, are not implied in the article. From the article, you can extract information about the features of hemorrhagic complications and their treatment during VV-ECMO patients with COVID-19.

A quantitative analysis of hemorrhagic complications and their treatment was carried out, suggestions were made about the causes of the features of VV-ECMO patients with COVID-19, including a possible connection with comorbid conditions.

The authors properly reported on their experience and, in my opinion, the article does not require any improvement. Another thing is whether the nature and content of this article correspond to the scope of the journal and the expectations of the Editor.

There is no need to talk about conclusions, evidence and arguments. The article reports the results of VV-ECMO in 114 patients with COVID-19 associated ARDS, while giving a quantitative analysis of the types of hemorrhagic complications (major, minor), their localization, volumes of blood transfusion

There are no flaws or drawbacks about the tables and figures, they fully correspond to the content of the article and improve the perception of the material.

Author Response

We would like to express our gratitude to the reviewer for the time and care taken and hope to further improve our manuscript by addressing the issue raised.

- Lines 102 and 105 indicate the concentration of hemoglobin in mg / dL, it should be g / dL

We thank the reviewer for the careful observation and have made an appropriate adjustment.

Round 2

Reviewer 2 Report

The specifications introduced in the methods improve the quality of the article. In relation to what has been written, it would be appropriate to report in table 1 the home use of anticoagulant and antiplatelet drugs.
Furthermore, I would ask the authors to specify in how many cases the use of the multiplex® analyzer (Roche, Basel, Swiss) was required and how this test changed their clinical and therapeutic approach

Author Response

We would like to thank the reviewer again for his attentive and competent review and hope that our explanations and corresponding extensions do credit to the reviewer's valuable objections.

For point at point editing please see the attachment.
